# A Comparison of Different Protocols for the Extraction of Microbial DNA Inhabiting Synthetic Mars Simulant Soil

**DOI:** 10.3390/microorganisms12040760

**Published:** 2024-04-10

**Authors:** Han Wang, Agata Pijl, Binbin Liu, Wieger Wamelink, Gerard W. Korthals, Ohana Y. A. Costa, Eiko E. Kuramae

**Affiliations:** 1Department of Microbial Ecology, Netherlands Institute of Ecology (NIOO-KNAW), 6708 PB Wageningen, The Netherlands; h.wang@nioo.knaw.nl (H.W.); a.pijl@nioo.knaw.nl (A.P.); o.costa@nioo.knaw.nl (O.Y.A.C.); 2Ecology and Biodiversity, Institute of Environmental Biology, Utrecht University, 3584 CH Utrecht, The Netherlands; 3Center for Agricultural Resources Research, Institute of Genetics and Developmental Biology, Chinese Academy of Sciences, 286 Huaizhong Road, Shijiazhuang 050021, China; binbinliu@sjziam.ac.cn; 4Biodiversity and Policy, Wageningen University and Research, Droevendaalsesteeg 4, 6708 PB Wageningen, The Netherlands; wieger.wamelink@wur.nl; 5Bioindications and Plant Health, Wageningen University and Research, Droevendaalsesteeg 4, 6708 PB Wageningen, The Netherlands; gerard.korthals@wur.nl

**Keywords:** MGS-1, FastDNA SPIN Kit, DNeasy PowerSoil Pro Kit, bacteria community

## Abstract

Compared with typical Earth soil, Martian soil and Mars simulant soils have distinct properties, including pH > 8.0 and high contents of silicates, iron-rich minerals, sulfates, and metal oxides. This unique soil matrix poses a major challenge for extracting microbial DNA. In particular, mineral adsorption and the generation of destructive hydroxyl radicals through cationic redox cycling may interfere with DNA extraction. This study evaluated different protocols for extracting microbial DNA from Mars Global Simulant (MGS-1), a Mars simulant soil. Two commercial kits were tested: the FastDNA SPIN Kit for soil (“MP kit”) and the DNeasy PowerSoil Pro Kit (“PowerSoil kit”). MGS-1 was incubated with living soil for five weeks, and DNA was extracted from aliquots using the kits. After extraction, the DNA was quantified with a NanoDrop spectrophotometer and used as the template for 16S rRNA gene amplicon sequencing and qPCR. The MP kit was the most efficient, yielding approximately four times more DNA than the PowerSoil kit. DNA extracted using the MP kit with 0.5 g soil resulted in 28,642–37,805 16S rRNA gene sequence reads and 30,380–42,070 16S rRNA gene copies, whereas the 16S rRNA gene could not be amplified from DNA extracted using the PowerSoil kit. We suggest that the FastDNA SPIN Kit is the best option for studying microbial communities in Mars simulant soils.

## 1. Introduction

Information from ground-based landers/rovers and orbital spacecraft indicates that the Martian surface is mainly covered with basaltic soil. This soil is primarily composed of pyroxene, plagioclase feldspar, and olivine, with small quantities of iron and titanium oxides, such as magnetite, ilmenite, and hematite, and alteration minerals, such as sulfates, phyllosilicates, and carbonates [1,2,3,4,5]. Numerous studies have suggested the potential existence of life, particularly microorganisms, on Mars, but no conclusive evidence is present [6,7,8]. In the absence of actual samples of Martian soil, Mars simulant soils are commonly used in laboratories on Earth to calibrate equipment, test rover movements, and simulate Mars exploration scenarios [9,10,11,12]. In addition, many educational institutions use Mars simulant soils for demonstrations on Martian geology and extraterrestrial agriculture [13,14,15]. Furthermore, recent research demonstrates that microorganisms can effectively enhance soil nutrients available and facilitate crop growth in Mars simulant soil [16,17,18]. This highlights the importance of obtaining soil DNA to understand the interaction between microorganisms and Mars simulant soils, and to perform subsequent sequencing analysis of microbial functions.

The high contents of strongly adsorbing components such as silicates, iron-rich minerals, sulfates, and metal oxides, which can also generate destructive hydroxyl radicals through cationic redox cycling, pose challenges for experimental procedures involving Mars simulant soils, such as DNA extraction [19,20,21]. Studies of the efficiency and reproducibility of DNA extraction from Mars simulant soils using commercial DNA extraction kits are scarce. In this study, we investigated the performance of DNA extraction methods using microbial communities assembled in the sterile Mars Global Simulant (MGS-1). This microbial community was obtained by adding MGS-1 into mesh bags and burying them in forest soil for five weeks, allowing the microbial community from the forest soil to migrate and assemble in the sterile Mars simulant soil. MGS-1 is a Mars simulant soil with pH 8.0 that contains abundant silicates, iron-rich minerals, sulfates, and metal oxides. The forest environment, characterized by its stability and rich microbial diversity, provides an ideal setting for exploring microbial assembly, colonization, and adaptation to the Mars simulant soil. We evaluated the effectiveness of two commercial kits, the FastDNA SPIN Kit for soil and the DNeasy PowerSoil Pro Kit for microbial DNA extraction from MGS-1. Finally, we quantified the abundance of the microbial community in the extracted DNA using the quantitative real-time PCR (qPCR) approach and determined the bacterial community composition and diversity using high-throughput 16S rRNA partial gene amplicon sequencing.

## 2. Material and Methods

### 2.1. Soil Sampling

Four topsoil (0–5 cm) samples were collected from the Wolfheze forest in the Netherlands (51°59′14.5″ N, 5°47′32.7″ E) in the spring of 2017. The samples were pooled, sieved (4 mm mesh), and used as the microbiome inoculum. 

MGS-1 was chosen as the Mars simulant soil because of its physical and chemical similarity to Martian soil. MGS-1 is an open-standard simulant designed as a high-fidelity mineralogical analog of global basaltic regolith on Mars, as represented by the Rocknest windblown deposit at Gale Crater [13]. MGS-1 is prepared by mixing terrestrial minerals in specific proportions. The chemical composition of MGS-1 is compared to that of Mars soil in Table 1. 

MGS-1 was sterilized by gamma irradiation (>35 kGy; Isotron, Ede, The Netherlands), which minimizes the effects on abiotic soil properties compared to autoclaving and freezing. Sterility was confirmed by spreading 0.5 g of sterilized soil onto 1/10 tryptic soy agar (TSA, 5 g L^−1^ NaCl, 1 g L^−1^ KH_2_PO_4_, 3 g L^−1^ tryptic soy broth, 20 g L^−1^ agar, pH 6.5) and potato dextrose agar (PDA; Oxoid, Hampshire, UK) and incubating the plates for six weeks. No bacterial or fungal growth was observed on the agar plates after incubation. 

### 2.2. Soil Sample Preparation

The moisture content and water holding capacity (WHC) of the forest soil and MGS-1 were measured prior to the laboratory inoculation experiment. WHC is defined as the ability of a soil matrix to hold water, controlled and heavily affected by soil organic matter and texture [22]. To determine WHC, 5 g of soil sample was deposited on a funnel with filter paper and homogeneously saturated with water. The extra water was drained from the bottom of the funnel. It was allowed to stand for a while. When the water drops stopped, the soil was removed and weighed immediately. Afterward, the soil was dried in an oven for 24 h at 105 °C and then weighed. To calculate the 100% WHC, the weight of the dry soil was subtracted from the weight of the wet soil, divided by the weight of the dry soil, and multiplied by 100. The forest soil was adjusted to 60% WHC, and three replicates of 200 g of soil were placed in 250 mL jars. The jars were covered with aluminum foil to block light and preincubated for one week in a climate room maintained at 75% relative humidity, 21 °C at night, and 25 °C in the day with a 16 h photoperiod. After the first week, 2 g of forest soil was collected for DNA extraction, and a 20 μm mesh nylon bag enclosing 45 g of sterile MGS-1 (60% WHC) was buried in each jar, ensuring that the bag was covered by the forest soil. The jars were then incubated for five weeks under the same conditions as in the first week. The moisture level in the jars was maintained at 60% WHC throughout the experiment. After the five-week period, the MGS-1 inside the nylon bags was collected and homogenized. The soil of each replicate was stored in aliquots of 10 g at −20 °C until further processing of DNA extraction, quantification, amplification, and sequencing (Figure 1).

### 2.3. Soil DNA Extraction 

The total soil DNA was extracted from MGS-1 using the DNeasy^®^ PowerSoil^®^ Pro Kit (Qiagen, Mo Bio Laboratories, Hilden, Germany) and FastDNA^®^ SPIN Kit (MP Biomedicals, Eschwege, Germany) according to each manufacturer’s protocol, with specific adjustments detailed below. The total soil DNA was extracted from forest soil using the optimal kit for MGS-1. The final DNA concentration and purity were determined in a NanoDrop 2000 UV–vis spectrophotometer (Thermo Scientific, Wilmington, NC, USA), and DNA quality was checked by 1% agarose gel electrophoresis. 

#### 2.3.1. PowerSoil DNA Isolation Kit

The PowerSoil DNA Isolation Kit (hereafter “PowerSoil”) was used to extract total DNA from MGS-1 according to the manufacturer’s protocol in triplicate with the following adjustments:

Normal method: The total DNA was extracted from 500 mg of MGS-1 according to the manufacturer’s protocol without modification. Briefly, we added 500 mg of soil and 800 μL of Solution CD1 in the PowerBead Pro Tube and homogenized the mixture in TissueLyser II (Qiagen, Hilden, Germany) for 10 min at 30 s^−1^ frequency speed. We centrifuged the PowerBead Pro Tube at 15,000× *g* for 1 min, and the supernatant was transferred to a new tube. Next, 200 μL of Solution CD2 was added, mixed briefly, and centrifuged again at 15,000× *g* for 1 min. The resulting supernatant was transferred to another new tube, and 600 μL of Solution CD3 was added, mixed briefly, and then loaded onto an MB Spin Column. After centrifugation at 15,000× *g* for 1 min, the flow-through was discarded, and the column was washed with Solution EA and then Solution C5. Finally, the DNA was eluted from the column using 50 μL Solution C6 from the kit and stored at −20 °C.

Strong beating method: MGS-1 (500 mg) was mixed with 800 μL of Solution CD1 in the PowerBead Pro Tube and homogenized in TissueLyser II (Qiagen, Hilden, Germany) for 8 min at maximum speed. This homogenization step was repeated three times, with rest intervals of 2 min on ice. The next steps followed the manufacturer’s protocol.

Heating method: MGS-1 (500 mg) was mixed with 800 μL of Solution CD1 in the PowerBead Pro Tube and heated for 30 min at 65 °C. The next steps followed the manufacturer’s protocol.

Cation exchange resin (CER) method [23,24]: MGS-1 (500 mg) was mixed with 800 μL of Solution CD1 and 250 mg of CER (Dowex ‘Marathon C’ sodium form, strongly acidic, 20–50 mesh) in the PowerBead Pro Tube. The next steps followed the manufacturer’s protocol.

Phosphate-buffered saline (PBS) method [25]: MGS-1 (2 g) was mixed with 5 mL of PBS (137 mM NaCl, 2.7 mM KCl, 10 mM Na_2_HPO_4_, and 1.8 mM KH_2_PO_4_ adjusted to a final pH of 7.2) with shaking for 30 min at 4 °C. The mixture was centrifuged at 8000× *g* for 10 min, and the supernatant was discarded. The pellet was suspended in 800 μL of Solution CD1 and transferred to the PowerBead Pro Tube. The next steps followed the manufacturer’s protocol.

PBS and CER methods: MGS-1 (2 g) was mixed with 5 mL of PBS and 1 g of CER with shaking for 30 min at room temperature. The mixture was centrifuged at 15,000× *g* for 2 min, and the supernatant was discarded. The pellet was resuspended in 800 μL of Solution CD1 and transferred to the PowerBead Pro Tube. The next steps followed the manufacturer’s protocol.

Liquid nitrogen method [26,27]: MGS-1 (500 mg) was mixed with 500 μL of liquid nitrogen in the PowerBead Pro Tube and ground with a sterile pestle for 5 min. After the liquid nitrogen was volatilized, 800 μL of Solution CD1 was added to the PowerBead Pro Tube, and the mixture was heated for 30 min at 65 °C. The next steps followed the manufacturer’s protocol.

#### 2.3.2. FastDNA SPIN Kit 

In triplicate, 500 mg of MGS-1 was mixed with 978 μL of sodium phosphate buffer (a reagent in the kit) and 122 μL of MT Buffer in Lysing Matrix E tubes from the FastDNA SPIN Kit (hereafter “MP”). The mixture was homogenized in the FastPrep^®^ machine for 40 s at a speed setting of 6.0. This homogenization step was repeated twice, with a 5 min rest interval on ice. We followed the manufacturer’s protocol to perform all further steps involving proprietary reagents and spin filters. The DNA was eluted in 50 μL of DES reagent from the kit and stored at −20 °C.

### 2.4. High-Throughput Sequencing of Bacterial Community Composition

The DNA extracts were used in high-throughput 16S rRNA partial gene sequencing to assess bacterial community composition and diversity. The primer pair 338F (5′-ACTCCTACGGGAGGCAGCA-3′) and 806R (5′-GGACTACHVGGGTWTCTAAT-3′) was used to amplify the V3-V4 hypervariable region of the bacterial 16S rRNA gene for amplicon sequencing. Polymerase chain reaction (PCR) targeting the 16S rRNA region was performed in 25 μL reactions containing 12.5 μL of PCR premix (Phanta Max Super-Fidelity DNA Polymerase, Vazyme Biotech Co., Ltd., Nanjing, China), 1 μL of each primer (10 μM), and 1 μL of DNA template (approximately 20 ng of DNA). The PCR conditions consisted of an initial denaturation at 95 °C for 3 min, followed by 25 cycles of 30 s at 95 °C, 30 s at 55 °C, and 30 s at 72 °C, with a final extension at 72 °C for 10 min. The PCR products were analyzed by separation using 2% agarose gel electrophoresis and purified with AMPure XP beads (Beckman Coulter, Inc., Brea, CA, USA) according to the manufacturer’s instructions. Subsequently, eight-cycle PCR was performed to attach dual-index barcodes and Illumina sequencing adapters to each sample, followed by the purification of the PCR products using AMPure beads. Equimolar amounts of the PCR products were sequenced using the Illumina MiSeq PE300 platform (Illumina, Inc., San Diego, CA, USA) at Allwegene Technology Inc. Co., Ltd. (Beijing, China). The datasets supporting the conclusions of this article are available in the European Nucleotide Archive (ENA; https://www.ebi.ac.uk/ena, 20 February 2024) under accession number PRJEB68335.

### 2.5. Sequence Read Processing and Data Analysis

Adapter and primer sequences were removed with Cutadapt v2020.11.1 [28]. The forward and reverse reads were processed using DADA2 within QIIME2 v2023-02 to identify amplicon sequence variants (ASVs) [29,30]. Quality trimming, denoising, merging, and chimera detection were performed using the QIIME2 v2023.02 plugin “qiime dada2 denoise-paired”. Default settings were applied, except for “--p-trunc-len-f” and “--p-trunc-len-r”, which were set at 228 and 224, respectively. The taxonomic lineage of representative sequences derived from the ASVs was classified using the “classify-sklearn” plugin [31]. This classification was performed against the SILVA database (v. 138) for bacteria [32]. The ASV tables were converted into tab-separated value (tsv) format and exported using the BIOM package (version 1.30.0) [33].

All statistical analyses were performed using R 4.3.1 with different packages. To standardize sample sizes to the smallest number of non-chimeric sequences, each sample was rarefied to 26,160 reads. Rarefaction involves subsampling the sequencing data to control for uneven sequencing effort and make the sequencing depth consistent across all samples for fair comparisons [34]. To evaluate the stability of the relative abundances of individual taxa post-incubation, the coefficient of variation (CV) was utilized [35,36]. The CV was calculated by dividing the standard deviation by the mean for all samples and multiplying the result by 100.

### 2.6. Real-Time Quantitative PCR (qPCR)

The total abundance of bacteria present in DNA samples was quantified by performing the quantitative PCR (qPCR) of the 16S rRNA gene in a Real-Time PCR Detection System (Bio-Rad CFX96 Touch^TM^, Bio-Rad, Hercules, CA, USA). qPCR was performed in a total volume of 12 μL containing 5 μL of SYBR Green Bioline SensiFAST SYBR^®^ No-ROX mix (Bioline, London, UK), 0.125 μL of each primer (10 pmol), 2 μL of BSA, and 2 μL of DNA (20 ng); the primer pair used was EUB338 (5′-CTCCTACGGGAGGCAGCAG-3′) and EUB518 (5′-ATTACCGCGGCTGCTGG-3′). The thermal cycler conditions were 95 °C for 10 min; 40 cycles of 95 °C for 10 s, 53 °C for 10 s, and 72 °C for 20 s; acquisition was performed at 53 °C. The qPCR amplicon products (200 bp) were checked by melting curve analysis and agarose gel electrophoresis. The efficiency of the 16S rRNA gene qPCR was 96% (R^2^ = 0.99). Plasmid DNA containing fragments of the 16S rRNA gene was used as a standard. Each qPCR run was performed in triplicate and included a DNA template, the standard positive control, and a negative control.

## 3. Result

The total yields of extracted DNA from MGS-1 samples (FM) measured by NanoDrop analysis are presented in Table 2. Despite the various modifications, the yields of the extracted DNA from the PowerSoil kit were generally low. Moreover, the quality of the DNA was poor, with many A_260/280_ ratios < 1.8 and all A_260/230_ ratios < 2.0. Furthermore, no amplicons were obtained by PCR or qPCR using the extracted DNA as a template.

By contrast, DNA extraction using the MP kit was successful. Although the DNA yields were not exceptionally high, the DNA yield and quality were stable. The A_260/280_ and A_260/230_ ratios of the extracted DNA were relatively high and low, respectively, indicating the presence of RNA; high salt concentrations; and contaminants such as EDTA, carbohydrates, and phenol. Nonetheless, the extracted DNA was successfully used in Illumina high-throughput sequencing. The number of reads generated by 16S rRNA amplicon sequencing ranged from 28,642 to 37,805, and the gene copy numbers determined by qPCR ranged from 30,380 to 42,070. In FASTQ format, there were 228 sequence bases in the forward sequence and 224 sequence bases in the reverse sequence, all with high-quality scores between 25 and 40. The analysis of bacterial community composition based on 16S rRNA gene amplicons showed that the most abundant phyla in the DNA samples extracted with the MP kit were *Cyanobacteriota*, *Pseudomonadota*, and *Actinomycetota* (48.95%, 15.78%, and 13.62%, respectively). This contrasts with a shift observed in the relative abundance of *Cyanobacteriota*, *Pseudomonadota*, and *Actinomycetota* (0.26%, 33.98%, and 22.64%, respectively) in the DNA samples extracted from the original forest soil using the same MP kit (Figure 2).

## 4. Discussion

The high silicate, iron-rich mineral, sulfate, and metal oxide content of Mars simulant soil and its relatively high pH (8.0) present challenges for DNA [19,37,38]. DNA binding to cations and minerals and degradation caused by hydroxyl radicals at high pH can further complicate extraction, especially when microbial biomass is low [20]. In this study, we evaluated commercial DNA extraction kits with the goal of selecting the most suitable kit for Mars simulant soils based on DNA yield and purity, amplicon sequencing, and the qPCR amplification of the 16S rRNA bacterial gene.

We implemented several modifications to the PowerSoil kit for soil DNA extraction in an attempt to enhance extraction efficiency. These adjustments included prolonging the bead-beating time, applying heat to facilitate cell lysis and DNA release, using PBS for soil washing, incorporating CER to adsorb metal cations and release DNA, using a combination of PBS and CER for soil washing to eliminate impurities and salt effects, and employing liquid nitrogen to grind the soil for DNA release. None of these modifications resulted in successful bacterial 16S rRNA partial gene amplification by qPCR, PCR, and sequencing.

Unlike the PowerSoil kit, the MP kit successfully extracted DNA from MGS-1. The DNA extracted using the MP kit was successfully used in 16S rRNA gene amplicon sequencing to obtain ASVs. These findings were further substantiated by the qPCR results. Previous studies have shown that the MP kit provides considerably higher DNA yields from the soil than other DNA isolation kits, including the Power Soil Kit, albeit at the cost of high humic acid contamination [39,40]. Humic acid is not expected to be present in Mars simulant soils, although the high amount of salt may reduce DNA quality. Furthermore, we were also able to extract DNA by using the MP kit from another Mars simulant soil (Mars Regolith Simulant, MMS-1) [41] after 8-week incubation with a bacterial inoculum, resulting in a DNA yield of approximately 23.5 ng/μL, A_260/280_ ratio of 1.75, A_260/230_ ratio of 0.11, and 7 × 10^10^ g^−1^ 16S rRNA gene copies. MMS-1 is a basalt that is mined from the Tertiary Tropico Group in the Western Mojave Desert and crushed to gravel, dust, and sand grades, and it has a higher pH (9.0) than MGS-1 (pH 8.0). Overall, the MP kit effectively extracted DNA from MGS-1, a Mars simulant soil with very unusual chemical characteristics.

Because of the proprietary nature of the reagents and chemical mechanisms of the commercial extraction kits, we can only speculate on the reasons for the differences in their effectiveness. First, the PowerBead tubes in the PowerSoil kit contain guanidine salts, which may not be suitable for Mars simulant soils. By contrast, the Lysing Matrix E tubes in the MP kit contain a mixture of ceramic and silica particles that are designed to efficiently lyse all soil organisms. Second, we observed that the MGS-1 in the Lysing Matrix E tubes from the MP kit was finer and more uniform after the beating step than the MGS-1 in the PowerBead tubes from the PowerSoil kit. This difference might be attributable to the FastPrep^®^ Instrument, which produces rapid, efficient, and highly reproducible homogenization surpassing that obtained by traditional methods such as enzymatic digestion, sonication, blending, bouncing, and vortexing. The FastPrep^®^ Instrument employs a unique motion that facilitates sample homogenization through multidirectional, simultaneous impacts with the lysing matrix particles. Third, in the MP kit, DNA is purified from the supernatant with the Binding Matrix FastDNA procedure using SPIN filters. The combined application of physical force, appropriate compounds for efficient lysis, and the special binding matrix contribute to the effectiveness of the MP protocol with Mars simulant soils.

There are no perchlorates, chlorate salts, or other superoxide in MGS-1. However, it is notable that perchlorate was detected in Martian soil by instruments aboard both the Mars Phoenix Lander and the Mars Science Laboratory [42,43,44]. When extracting samples that may cause soil–DNA interactions such as perchlorates, basalt, and alkaline soil components, it may be helpful to use desalting and competitive binding agents, such as random hexamer primers and skim milk, during the cell lysis phase of the DNA extraction process. These methods are aimed at purging highly damaging cations and additional inhibitory ions [20].

Different studies have found that the effectiveness of DNA extraction kits varies depending on the type of Mars simulant soil, how microbes are added to the soil, and the research methods used [20,21,34,38,45]. Xia effectively used the PowerSoil kit and ZymoBiomics DNA Microprep Kit to extract DNA from five other Mars simulant soils that are collected from five different terrestrial deposits each. This success may be due to differences in the types of Mars simulant soils used and the method of directly extracting DNA without prior incubation, focusing on single bacterial species inoculation [38]. It is important to note that we used a complete soil microbial community. The effectiveness of the MP and PowerSoil kits varies across different Mars simulant soil samples collected from various geological formations as well. PowerSoil kits proved more effective in Morrison samples, pH 8.11–10.1, belonging to the late Jurassic period. The MP kit showed more effectiveness in Mancos Shale samples, which are from the Cretaceous period, with pH 7.5–8.4 [45]. However, those Mars simulant soils are often sourced from a single terrestrial deposit, typically only one or two mineral components. MGS-1 is made by sourcing individual minerals including a proper treatment of the X-ray amorphous component. This is in contrast with other Mars simulant soils.

This study highlights the complexity of extracting DNA from Mars simulant soil MGS-1, which has a pH of 8.0 and is abundant in silicates, iron-rich minerals, sulfates, and metal oxides. The MP kit not only overcomes the challenges posed by these soil properties but also provides accurate, reliable, and repeatable results.

## Figures and Tables

**Figure 1 microorganisms-12-00760-f001:**
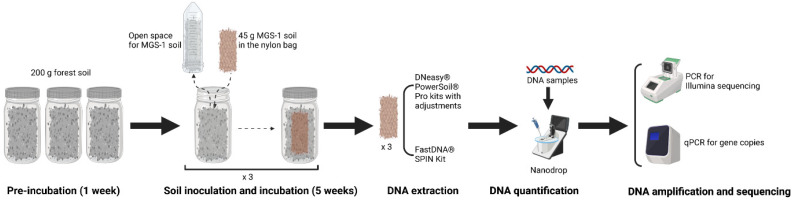
Schematic illustration of inoculation of MGS-1 and DNA extraction, quantification, amplification, and sequencing.

**Figure 2 microorganisms-12-00760-f002:**
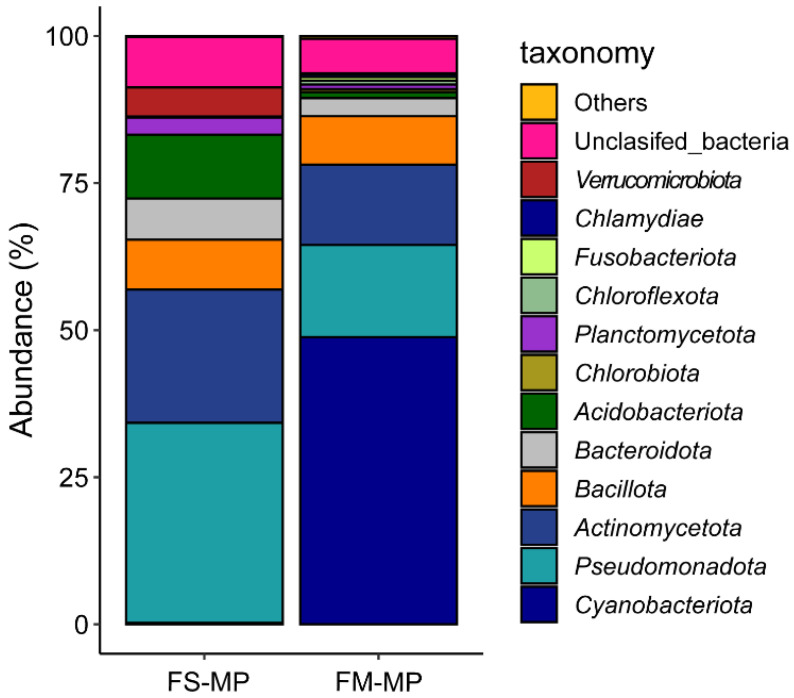
Bacterial composition of forest soil DNA samples (FS-MP) and MGS-1 DNA samples (FM-MP) extracted with the MP kit based on relative abundances at the phylum level. Phyla with relative abundances of less than 0.1% were classified together as “Others”.

**Table 1 microorganisms-12-00760-t001:** Percentage of the chemical composition of Mars Global Simulant (MGS-1) and Mars soil.

ID	MGS-1	Mars
SiO_2_	45.57%	43.52%
Fe_2_O_3_	16.85%	18.28%
Al_2_O_3_	9.43%	8.64%
CaO	4.03%	6.09%
MgO	16.50%	6.54%
SO_3_	2.63%	6.42%
Na_2_O	3.66%	2.57%
P_2_O_5_	0.37%	0.79%
TiO_2_	0.30%	0.78%
K_2_O	0.43%	0.35%
MnO	0.10%	0.32%
Cr_2_O_3_	0.12%	0.37%
Total	99.99%	94.67%

**Table 2 microorganisms-12-00760-t002:** DNA yield, A_260/280_ ratio, A_260/230_ ratio, number of 16S rRNA amplicon sequencing reads, and 16S rRNA gene copies/g from each soil DNA extraction method.

	Sample ID	DNA Yield (ng/µL)	A_260/280_ *	A_260/230_ *	Method	Number of Reads	16S rRNA Gene Copies/g
	FM-1	1.8	1.74	0.23	Normal	0	0
	FM-2	1.8	1.11	0.33	Normal	0	0
	FM-3	3.2	1.2	0.17	Normal	0	0
	FM-B1	68.8	1.55	0.4	Beating	0	0
	FM-B2	12.5	1.61	0.18	Beating	0	0
	FM-B3	68.8	1.55	0.4	Beating	0	0
	FM-H1	0.5	1.91	0.05	Heating	0	0
	FM-H2	1.6	1.21	0.08	Heating	0	0
PowerSoil	FM-H3	1.2	1.56	0.01	Heating	0	0
	FM-CER1	5	0.75	0.03	CER	0	0
	FM-CER2	2.4	2.25	0.02	CER	0	0
	FM-CER3	6	0.87	0.06	CER	0	0
	FM-P1	9	1.62	0.03	PBS	0	0
	FM-P2	13	1.89	0.13	PBS	0	0
	FM-P3	9	3.15	0.03	PBS	0	0
	FM-PC1	5	5.31	0.04	PBS+CER	0	0
	FM-PC2	9	1.28	0.01	PBS+CER	0	0
	FM-PC3	19	1.39	0.01	PBS+CER	0	0
	FM-N1	1.3	1.59	0.04	N2	0	0
	FM-N2	1.1	1.93	0.03	N2	0	0
	FM-N3	3.2	1.47	0.08	N2	0	0
	FM-MP1	9.144	2.47	0.01	Normal	28,642	30,380
MP	FM-MP2	9.331	2.7	0.01	Normal	29,743	38,550
	FM-MP3	8.391	2.62	0.01	Normal	37,805	42,070

* A_260/280_, the ratio of absorbance at 260 nm and 280 nm is used to assess the purity of DNA. A_260_: DNA absorbs with a peak at 260 nm. A_280_: protein absorbs with a peak at 280 nm. A_260/230_: the ratio of absorbance at 260 nm and 230 nm is used as a secondary measure of DNA purity. A_260/280_ around 1.8 and A_260/230_ in the range of 2.0–2.1 means the DNA is generally accepted as “pure”.

## Data Availability

The datasets supporting the conclusions of this article are available in the European Nucleotide Archive (ENA; https://www.ebi.ac.uk/ena, accessed on 20 February 2024) under accession number PRJEB68335.

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
