# Peer review of "A Comparison of Different Protocols for the Extraction of Microbial DNA Inhabiting Synthetic Mars Simulant Soil"

_microorganisms, 2024, doi:10.3390/microorganisms12040760_

Round 1
Reviewer 1 Report
Comments and Suggestions for Authors
The manuscript is testing different methods to extract DNA from Mars simulant soil.
The addressed topic is relevant. However, the authors did not take into consideration the presence of perchlorate in the Martian soils (doi:10.1017/S1473550415000385; doi:10.1017/S1473550416000057 ).
In addition, since the authors used a “Heating method” to improve the DNA release, it should be taken into consideration that perchlorates are powerful oxidants when heated(10.1089/ast.2013.0999).
Therefore these two aspects should be discussed in the paper.
Author Response
"Please see the attachment."

Reviewer 2 Report
Comments and Suggestions for Authors
The authors test two types of putative DNA extraction methods from a Martian simulant, and show that one performs better than the other. However, there are a number of questions and improvements that the authors should address first.
1. In lines 35-38, the authors should mention perchlorates and the challenges posed by this class of compounds to habitability.
2. As noted by the authors in the Abstract, the objective of this work is to address the challenges of "extracting microbial DNA" from Martian soil. However, this statement is relevant only if Mars has/had microbial life in the first place. Yet, there is no explicit discussion of Martian habitability and life in the Introduction. Therefore, please add this topic, and the authors are suggested to cite all the following references:
https://www.liebertpub.com/doi/10.1089%2Fast.2013.1106
https://www.liebertpub.com/doi/full/10.1089/ast.2015.1374
https://www.liebertpub.com/doi/full/10.1089/ast.2016.1533
https://agupubs.onlinelibrary.wiley.com/doi/full/10.1029/2017JE005478
https://www.liebertpub.com/doi/full/10.1089/ast.2020.2237
https://www.hup.harvard.edu/books/9780674987579
https://www.cambridge.org/core/journals/international-journal-of-astrobiology/article/mars-new-insights-and-unresolved-questions/F0E43D7EC62EA126262CB66DF069ABA0
3. Why was the Wolfheze forest chosen? It does not seem a good Mars analog. Surely, microbes from an environment like Atacama or Antarctica would be much better. The authors should include careful justification of their choice of biome.
4. Please define (and/or give formulae for) all terms for the broad audience of Life, such as WHC, A260/280, etc.
5. As per Fig. 1, is the sample actually a mixture of the forest soil and MGS-1? If this is correct, then the authors cannot claim to demonstrate DNA extraction from a Mars-analog soil, because forest soil is not Mars-like.
6. Please provide more details about the various methods in Section 2.3.1. They are too brief, and thus confusing for non-specialists.
7. Lines 170-172 are not clear. Are the authors stating that they are excluding some data?
8. What is the difference between Section 2.4 and 2.6? Is there a difference in the method, or is it connected to the stages where PCR is employed? It is not sufficiently evident, since the methods seem similar.
9. Lines 197-198 appear to be unjustified. A260/280 ratios for some of the PowerSoil cases are higher than MP, which makes the former better; this statement, however, contradicts what the authors write here and elsewhere (e.g., in Section 4).
10. Authors should comment on what the inclusion of perchlorates would do to their findings.
11. The authors show that MP kit might be more well-suited for DNA extraction from Mars soils, but they should also discuss what other alternatives exist, and how they can be deployed. Moreover, a big caveat is that the authors perform their studies at temperatures, pressures, radiation, etc. very different from that of the actual Mars.
Comments on the Quality of English LanguageEnglish language is okay, but some of the statements are too brief or unclear and would benefit from proofreading and rewriting.
Author Response
"Please see the attachment."

Round 2
Reviewer 1 Report
Comments and Suggestions for Authors
I don't have any further comments.
Reviewer 2 Report
Comments and Suggestions for Authors
The authors have responded to my questions, and mostly addressed my concerns. The paper is acceptable for publication.